# OpenReview forum: "EXPLORING RESPONSE UNCERTAINTY IN MLLMS: AN EMPIRICAL EVALUATION UNDER MISLEADING SCENARIOS"
_ICLR.cc/2025/Conference — Submitted to ICLR 2025_

### Official Review · Reviewer_LrHb · 2024-10-19

**Soundness:** 2
**Presentation:** 3
**Contribution:** 3
**Rating:** 6
**Confidence:** 4

**Summary:**

This paper addresses the issue of response uncertainty in Multimodal Large Language Models (MLLMs), which can be problematic when models encounter misleading information. To tackle this, the authors propose a two-stage pipeline: first, gathering MLLM responses without misleading information, followed by collecting responses influenced by specific misleading instructions. They effectively evaluate model uncertainty by measuring the misleading rate and tracking shifts between correct and incorrect responses. They introduce the Multimodal Uncertainty Benchmark (MUB), which uses explicit and implicit misleading instructions to assess MLLM vulnerability across various domains. To improve robustness, they fine-tune open-source MLLMs using misleading data, substantially reducing misleading rates.

**Strengths:**

The strengths of this paper include:

- This work focuses on an important issue: the robustness of Multimodal Large Language Models (MLLMs) when faced with misleading instructions. This is a compelling research topic that addresses a gap in the current field.

- The paper is well-structured, with a clear framework, and the authors present three research questions that are thoroughly examined through extensive experiments involving 12 models.

- This work contributes to the community by introducing the Multimodal Uncertainty Benchmark and providing a fine-tuning dataset, demonstrating improved model robustness against misleading instructions.

**Weaknesses:**

The weaknesses of this work include:
- The paper lacks a discussion on whether the models are calibrated. It does not address whether more consistent (more certain) model outputs correspond to more accurate answers. The results in the paper are primarily based on misleading rate (MR) and average consistency rate (ACR), without showing metrics like model accuracy. There is a lack of analysis on utility.
- The authors fail to analyze the impact of instruction tuning on model usability. It is unclear how much the model’s performance on different tasks changes before and after fine-tuning. This lack of explanation limits the understanding of the benchmark’s functionality. Researchers are left uncertain whether fine-tuning causes models to generate consistent but incorrect responses.

Suggestions:
The images in the paper are quite blurry, especially Figure 2. The authors should check the image quality. There are also some typos, such as mixed-use of InternVL-Chat-V1-5  and Internvl-chat-v1.5.

**Questions:**

Refer to Weakness. The analysis of utility and calibration is important for such work.

---

> ### Author Response · Authors · 2024-11-20
>
> We greatly appreciate the thorough evaluation of our work and the valuable suggestions provided by the reviewers. Your feedback has been instrumental in helping us improve the clarity and comprehensiveness of our manuscript. In response to your concern, we conducted additional experiments and revised our analysis accordingly.
>
> > Q1: The paper lacks a discussion on whether the models are calibrated.
>
> A1: Thank you for your valuable suggestion. We have added a discussion in the appendix to address the calibration of the models. This discussion provides an analysis from three key perspectives:
> - **ECE.** We adopted the method described in [1] to enable the MLLMs to output their confidence scores. The results demonstrate a notable improvement in calibration, with the Expected Calibration Error (ECE) in the True-False (T-F) scenario decreasing from 0.47 to 0.23 after fine-tuning.
> - **Consistency Rate.** The consistency rate of the fine-tuned MLLMs shows significant improvement. Specifically, we observed an approximate 13.58% increase in low misleading rate data and a 26.51% increase in high misleading rate data, as detailed in the revised Table 21.
> - **Accuracy.** The accuracy of the fine-tuned MLLMs also improved moderately, with an approximately 5% increase observed after fine-tuning. These results are reported in the updated Tables 17 and 18 in the revised manuscript.
>
>
> #### Table: Comparison of ECE before and after fine-tuning on our benchmark
> | Model                  | Before | After |
> |------------------------|--------|-------|
> | MiniCPM-v-v2           | 0.46   | 0.24  |
> | Phi-3-vision           | 0.46   | 0.15  |
> | Yi-VL-6b               | 0.45   | 0.27  |
> | Qwen-VL-Chat           | 0.49   | 0.24  |
> | Deepseek-VL-7b-Chat    | 0.47   | 0.20  |
> | LLaVA-NeXT-7b-vicuna   | 0.48   | 0.23  |
> | MiniCPM-Llama3-v2.5    | 0.49   | 0.18  |
> | GLM4V-9B-chat          | 0.46   | 0.25  |
> | CogVLLM-chat           | 0.46   | 0.27  |
> | InternVL-Chat-V1-5     | 0.47   | 0.24  |
> | LLaVA-Next-34b         | 0.49   | 0.19  |
> | Yi-VL-34b              | 0.45   | 0.26  |
> | **Average**            | **0.47** | **0.23** |
>
>
> #### Table: Comparison of consistency before and after fine-tuning on our benchmark
> | **Model**         | **Low (Before)** | **Low (After)** | **Low (Change)** | **High (Before)** | **High (After)** | **High (Change)** |
> |--------------------|------------------|-----------------|------------------|-------------------|------------------|-------------------|
> | MiniCPM-v-v2      | 82.93%           | 97.83%          | +14.90%          | 56.52%            | 90.64%           | +34.12%           |
> | Phi-3-vision      | 79.89%           | 89.33%          | +9.44%           | 63.94%            | 87.77%           | +23.83%           |
> | GLM4v-9b          | 94.33%           | 99.00%          | +4.67%           | 82.28%            | 95.85%           | +13.57%           |
> | LLaVA-Next-34b    | 73.30%           | 98.61%          | +25.31%          | 53.30%            | 91.81%           | +38.51%           |
> | **Average**       | **82.61%**       | **96.19%**      | **+13.58%**      | **64.51%**        | **91.02%**       | **+26.51%**       |
>
> [1] Xiong, M.,et al. 2023. Can LLMs express their uncertainty? An empirical evaluation of confidence elicitation in LLMs.

---

> ### Author Response · Authors · 2024-11-20
>
> > Q2: It does not address whether more consistent (more certain) model outputs correspond to more accurate answers.The results in the paper are primarily based on misleading rate (MR) and average consistency rate (ACR), without showing metrics like model accuracy. There is a lack of analysis on utility.
>
> A2: Thank you for your valuable suggestion and for pointing out this important aspect of the analysis. We appreciate the opportunity to clarify and expand on this point. In the appendix of the original manuscript, we presented the changes in accuracy after fine-tuning in Table 22 on page 26 (updated as Table 17 and Table 18 in the revised version). To further address your concern and provide a clearer illustration of the fine-tuned model’s effectiveness, we have included the accuracy metrics directly in Table 2 of the main paper. We also provide the relationship between the accuracy and the misleading rate in Figure 10(in the revised version). The results indicate an inverse relationship between the misleading rate and the accuracy, where a higher misleading rate corresponds to a lower consistency rate. Also, we
>
>
> #### Table: the average accuracy before and after fine-tuning on our benchmark
> |                       | Explicit       |                  |                  | Implicit      |                  |                  |
> |--------------------|--------------|---------------------|------------------|------------------|---------------------|------------------|
> | Model       | Low                  | Medium           | High            | Low                |  Medium    | High      |
> | **Average** | **79.63% (↑4.95%)**  | **60.18% (↑5.10%)**  | **63.22% (↑5.69%)**  | **78.34% (↑4.07%)**  | **58.67% (↑4.76%)**  | **62.81% (↑5.55%)**  |
>
>
>
>
>
>
> ---

---

> > ### Author Response · Authors · 2024-11-20
> >
> > > Q3: The authors fail to analyze the impact of instruction tuning on model usability. It is unclear how much the model’s performance on different tasks changes before and after fine-tuning.
> >
> > A3: Thank you for your constructive feedback. As you recommended, we have provided additional experimental results on the SEED dataset (Table 20 in the new revision). The results demonstrate that fine-tuning leads to improved accuracy and a reduction in the misleading rate. Furthermore, we validated our approach on generative tasks, where the results indicate a significant decrease in the misleading rate for these tasks as well(Table 27 in the new revision). We also present the experimental results under various category divisions. The results indicate a substantial reduction in misleading rates across different tasks. (Table 24 in the new revision).
> >
> > #### Table: Comparison of accuracy and misleading rate before and after fine-tuning on SEED dataset
> > |                          | Before    |   |               | After       |   |             |
> > |--------------------------|------- |------- |--------|--------|--------|--------|
> > | Model                    | ACC    |  T-F   | F-T    | ACC    | T-F    | F-T    |
> > | MiniCPM-v-v2             | 63.65% | 53.45% | 87.02% | 71.00% | 6.76%  | 16.21% |
> > | Phi-3-vision             | 77.78% | 71.43% | 84.32% | 73.10% | 7.66%  | 27.88% |
> > | Yi-VL-6b                 | 60.26% | 83.73% | 96.59% | 69.80% | 15.62% | 27.15% |
> > | Qwen-VL-Chat             | 54.97% | 88.39% | 80.82% | 67.80% | 8.11%  | 17.08% |
> > | Deepseek-VL-7b-Chat      | 63.71% | 20.03% | 54.14% | 72.90% | 2.88%  | 4.80%  |
> > | LLaVA-NeXT-7b-vicuna     | 62.72% | 56.39% | 58.30% | 72.50% | 17.52% | 38.18% |
> > | MiniCPM-Llama3-v2.5      | 68.08% | 44.02% | 87.87% | 74.90% | 1.47%  | 1.20%  |
> > | GLM4V-9B-chat            | 68.71% | 32.93% | 78.03% | 75.20% | 4.12%  | 18.55% |
> > | CogVLLM-chat             | 67.73% | 24.69% | 65.96% | 75.60% | 8.20%  | 9.02%  |
> > | InternVL-Chat-V1-5       | 69.52% | 30.88% | 84.94% | 78.10% | 2.82%  | 4.11%  |
> > | LLaVA-Next-34b           | 67.40% | 41.07% | 95.06% | 76.50% | 2.09%  | 6.81%  |
> > | **Average**              | 66.44% | 51.72% | 78.47% | 73.00% | 7.47%  | 17.46% |
> >
> > #### Table: Comparison of explicit and implicit misleading instruction performance on generative tasks before and after fine-tuning
> > | Model                           | Before (T-F) | Before (F-T) | After (T-F) | After (F-T) |
> > |---------------------------------|--------------|--------------|-------------|-------------|
> > | **Explicit**                    |              |              |             |             |
> > | MiniCPM-v-v2                    | 69.23%       | 87.70%       | 25.00%      | 72.54%      |
> > | Phi-3-vision                    | 100.00%      | 66.67%       | 71.43%      | 30.57%      |
> > | Yi-VL-6b                        | 100.00%      | 82.89%       | 88.89%      | 55.50%      |
> > | Qwen-VL-Chat                    | 94.12%       | 86.34%       | 86.21%      | 50.88%      |
> > | Deepseek-VL-7b-Chat             | 92.31%       | 81.82%       | 70.59%      | 43.17%      |
> > | LLaVA-NeXT-7b-Vicuna            | 100.00%      | 62.56%       | 100.00%     | 60.20%      |
> > | MiniCPM-Llama3-v2.5             | 81.25%       | 83.71%       | 66.67%      | 64.29%      |
> > | GLM4V-9B-Chat                   | 85.71%       | 80.90%       | 48.48%      | 62.42%      |
> > | CogVLLM-Chat                    | 100.00%      | 54.55%       | 75.00%      | 3.35%       |
> > | InternVL-Chat-V1_5              | 85.71%       | 69.27%       | 24.32%      | 68.10%      |
> > | LLaVA-Next-34b                  | 100.00%      | 92.18%       | 62.50%      | 54.39%      |
> > | Yi-VL-34b                       | 90.91%       | 92.59%       | 77.78%      | 14.21%      |
> > | Average                         | 91.94%       | 76.99%       | 65.01%      | 48.31%      |
> > | **Implicit**                    |              |              |             |             |
> > | Average                         | 91.99%       | 44.38%       | 57.61%      | 23.57%      |

---

> > > ### Author Response · Authors · 2024-11-20
> > >
> > > #### Table: Comparison of explicit and implicit misleading instruction performance on different types of tasks before and after fine-tuning.
> > > | Model                  | Perception (T-F)     | Reasoning (T-F)      | Mastery (T-F)        |
> > > |------------------------|----------------------|----------------------|----------------------|
> > > | MiniCPM-v-v2           | 5.33% (↓78.37%)     | 7.28% (↓66.66%)      | 14.63% (↓59.73%)     |
> > > | Phi-3-vision           | 7.26% (↓78.62%)     | 6.62% (↓52.29%)      | 6.86% (↓56.46%)      |
> > > | Yi-VL-6b               | 9.42% (↓80.91%)     | 21.84% (↓66.49%)     | 46.92% (↓47.55%)     |
> > > | Qwen-VL-Chat           | 1.76% (↓90.06%)     | 7.78% (↓76.00%)      | 12.81% (↓68.33%)     |
> > > | Deepseek-VL-7b-Chat    | 1.42% (↓66.34%)     | 3.27% (↓54.71%)      | 6.78% (↓53.62%)      |
> > > | LLaVA-NeXT-7b-vicuna   | 4.81% (↓75.37%)     | 10.72% (↓44.31%)     | 15.68% (↓40.15%)     |
> > > | MiniCPM-Llama3-v2.5    | 0.73% (↓71.04%)     | 1.10% (↓63.18%)      | 1.75% (↓60.19%)      |
> > > | GLM4V-9B-chat          | 4.61% (↓39.82%)     | 8.39% (↓35.57%)      | 23.68% (↓35.80%)     |
> > > | CogVLLM-chat           | 8.13% (↓56.29%)     | 8.15% (↓37.65%)      | 32.40% (↓16.89%)     |
> > > | InternVL-Chat-V1-5     | 0.60% (↓49.74%)     | 2.85% (↓49.15%)      | 9.93% (↓50.51%)      |
> > > | LLaVA-Next-34b         | 2.12% (↓75.54%)     | 3.25% (↓84.97%)      | 2.25% (↓85.77%)      |
> > > | Yi-VL-34b              | 9.13% (↓71.55%)     | 17.12% (↓56.17%)     | 30.48% (↓37.84%)     |
> > > | Explicit Average       | 4.61% (↓69.47%)     | 8.20% (↓57.26%)      | 17.02% (↓51.07%)     |
> > >
> > >
> > > > Q4: This lack of explanation limits the understanding of the benchmark’s functionality. Researchers are left uncertain whether fine-tuning causes models to generate consistent but incorrect responses.
> > >
> > > A4: Thank you for highlighting this critical issue. We deeply appreciate your insightful observation. The fine-tuned model demonstrates no degradation in performance on our benchmark; in fact, achieving approximately a 5% performance improvement.  Moreover, its performance on other benchmarks, such as MMStar and AI2D, remains consistent, indicating that fine-tuning enables the model to generate more accurate responses. The results indicate a 22.49% improvement in accuracy after fine-tuning. Additionally, the consistency rate of the fine-tuned model shows a significant improvement to approximately 13.58% in the low misleading rate scenario and 26.51% in the high misleading scenario (in the below tables).  The fine-tuned models achieve an approximately 3% reduction in its average Expected Calibration Error (ECE), highlighting enhanced calibration and reliability.
> > >
> > >
> > > #### Table: The consistency rate of both before and after fine-tuning under low and high misleading rate scenarios.
> > > | Model                     | Low (Before) | Low (After) | Low (Change) | High (Before) | High (After) | High (Change) |
> > > |---------------------------|--------------|-------------|--------------|---------------|--------------|---------------|
> > > | MiniCPM-v-v2              | 82.93%       | 97.83%      | +14.90%      | 56.52%        | 90.64%       | +34.12%       |
> > > | Phi-3-vision              | 79.89%       | 89.33%      | +9.44%       | 63.94%        | 87.77%       | +23.83%       |
> > > | GLM4v-9b                  | 94.33%       | 99.00%      | +4.67%       | 82.28%        | 95.85%       | +13.57%       |
> > > | LLaVA-Next-34b            | 73.30%       | 98.61%      | +25.31%      | 53.30%        | 91.81%       | +38.51%       |
> > > | **Average**               | **82.61%**   | **96.19%**  | **+13.58%**  | **64.51%**    | **91.02%**   | **+26.51%**   |
> > >
> > >
> > > Table: The mean accuracy of 12 open-source models on the MMStar and AI2D datasets was evaluated both before and after fine-tuning under the high misleading rate scenario.
> > > | Model       | MMStar (Before) | MMStar (After) | AI2D (Before) | AI2D (After) |
> > > |-------------|-----------------|----------------|---------------|--------------|
> > > | **Average** | **44.02%**      | **45.67%**     | **66.60%**    | **67.94%**   |
> > >
> > >
> > > ---
> > > > Q5: The images in the paper are quite blurry, especially Figure 2. The authors should check the image quality. There are also some typos, such as mixed-use of InternVL-Chat-V1-5 and Internvl-chat-v1.5.
> > >
> > > A5: Thank you very much for pointing out this inconsistency. We have revised the figures and corrected typographical errors to ensure consistency and accuracy in the revised version.

---

> > > > ### Author Response · Authors · 2024-11-26
> > > >
> > > > Dear Reviewer LrHb,
> > > >
> > > > Thank you very much for your invaluable feedback on our paper. We have meticulously reviewed each of your points and endeavored to address them thoroughly. We would greatly appreciate it if you could review our responses and let us know if you have any additional comments regarding the paper or the rebuttal. We are eager to embrace all your critiques and integrate them into our work.
> > > >
> > > > Thank you!

---

> > > > > ### Comment · Reviewer_LrHb · 2024-11-26
> > > > >
> > > > > Thank you for your response. Your work addresses my concern and I've updated my rating.

---

### Official Review · Reviewer_zSd1 · 2024-11-01

**Soundness:** 2
**Presentation:** 4
**Contribution:** 2
**Rating:** 5
**Confidence:** 4

**Summary:**

This paper dives into how MLLMs often fail to perform well when faced with misleading prompts. To tackle this, the authors set up a benchmark called the Multimodal Uncertainty Benchmark (MUB), which first gathers standard responses and then throws in misleading inputs to see how often the models get tripped up. They then fine-tuned open-source models with a mix of straightforward and subtle misleading data, cutting down the rate of being misled, while keeping the models’ overall accuracy intact.

**Strengths:**

The attempt of addressing the response uncertainty in MLLMs is an interesting and important task. The proposed method seems largely sound to me in addressing at least parts of the problem. The paper is written in a structured and esay2understand manner -- quite straightforward. The MUB benchmark could be useful and the benchmarking results are generally informative. The effort of finetuning the MLLMs with misled data adds some more insights into how the problem could be mitigated.

**Weaknesses:**

My main concerns are

- This work only evaluates/tackles VLLM instead of MLLM as claimed multiple times in title and throughout paper, though I could maybe see the way to extend to other modalities.

- Having the implicit misleading information generated by GPT-4o seems like a "fighting fire with fire" approach -- I think it is better to have at least a subset of implicit ones written by human annotators so that we can see whether there is any difference between the human-generated ones and GPT-4o generated ones.

- During finetuning, a random set of explicit and implicit misled samples are used for finetuning, yet I am afraid the explicit misleading info has a too obvious and unique pattern due to how it's designed, hence too easy to pick them up, making the improvement after finetuning not too surprising.

- Instead of finetuning, I would recommend the authors to simply systematically prompt the MLLMs, such as "The questions might contain misleading information, you should try to answer the question correctly despite of those misleading information ..."; another version could even give it two examples (one explicit and one implicit). I would guess/assume, simply doing this extra prompting will make the results much better.

- The questions only include multi-choice and T/F styles, which certainly makes the metrics calculation easier (reflected in equation 1 and 2), yet probably losing the delicacy in the type of Q/A addressed?

**Questions:**

- The misleading information was only added to the textual questions, why not consider altering the image to inject misleading information?

---

> ### Author Response · Authors · 2024-11-20
>
> We sincerely thank for your thoughtful and detailed feedback, which has provided invaluable insights for improving our work. We appreciate the recognition of the importance of addressing response uncertainty in MLLMs and the potential utility of our proposed MUB benchmark. Your insights regarding the limitations and areas for improvement in our work are invaluable, and we have carefully addressed these concerns in the revised manuscript. Below, we provide detailed responses to each of the points raised.
>
> >Q1: We appreciate your valuable suggestions. We also present the additional experimental results in video question answering (video-QA). This work only evaluates/tackles VLLM instead of MLLM as claimed multiple times in title and throughout paper, though I could maybe see the way to extend to other modalities.
>
> A1:  Thank you for your valuable suggestions. In response, we have conducted additional experiments in the video question answering (Video-QA) setting to evaluate the adaptability of our method to multimodal inputs, including video and audio modalities. Specifically, we tested VideoLLaMA-2 [1] on the Video-MME [2] dataset across various categories. The results, shown in Table 28 and 29 (revised version), demonstrate that our method is effective for video and audio modalities. For the **video-audio modality**, the average accuracy decreased from 48.3% to 40.4%. Similarly, for models employing only the **video modality**, the average accuracy decreased from 54.9% to 45.5%.
>
> #### Table: Comparison of results before and after adding misleading instructions with video-audio input for VideoLLaMA-2 on the Video-MME dataset.
> | **Category**               | **Short Before** | **Short After** | **Medium Before** | **Medium After** | **Long Before** | **Long After** | **Overall Before** | **Overall After** |
> |----------------------------|------------------|-----------------|-------------------|------------------|-----------------|----------------|--------------------|-------------------|
> | **Knowledge**              | 59.6%           | **51.1%**      | 45.2%            | **38.5%**       | 39.3%          | **31.1%**     | 48.0%             | **40.2%**        |
> | **Film & Television**      | 68.3%           | **56.7%**      | 51.7%            | **43.3%**       | 35.8%          | **27.5%**     | 51.9%             | **42.5%**        |
> | **Sports Competition**     | 50.7%           | **43.3%**      | 44.7%            | **36.0%**       | 33.3%          | 31.3%         | 42.9%             | **36.9%**        |
> | **Artistic Performance**   | 61.7%           | **55.0%**      | 49.2%            | **44.2%**       | 44.2%          | **35.8%**     | 51.7%             | **45.0%**        |
> | **Life Record**            | 60.0%           | **51.0%**      | 43.3%            | **34.8%**       | 43.3%          | **34.8%**     | 48.9%             | **40.2%**        |
> | **Multilingual**           | 56.7%           | **36.7%**      | 36.7%            | **30.0%**       | 43.3%          | **26.7%**     | 45.6%             | **33.3%**        |
>
> [1] Cheng, Z., et al., 2024. VideoLLaMA 2: Advancing Spatial-Temporal Modeling and Audio Understanding in Video-LLMs.
>
> [2] Fu, C., et al., 2024. Video-MME: The first-ever comprehensive evaluation benchmark of multi-modal LLMs in video analysis.
>
> ---

---

> > ### Author Response · Authors · 2024-11-20
> >
> > > Q2: Having the implicit misleading information generated by GPT-4o seems like a "fighting fire with fire" approach -- I think it is better to have at least a subset of implicit ones written by human annotators so that we can see whether there is any difference between the human-generated ones and GPT-4o generated ones.
> >
> > A2: Thank you for your suggestion. We conducted additional experiments involving human-generated implicit misleading instructions and performed a comparative analysis with those generated by GPT-4o:
> > **Human-Generated Implicit Instructions:** We randomly selected 100 samples for annotation. Data annotators, all holding at least a bachelor's degree, created implicit misleading instructions based on the images, questions, options, and correct answers. The experimental results indicate that the misleading rates and levels of implicitness of human-generated instructions are comparable to those produced by GPT-4o. On average, approximately 4 minutes are required to generate a single implicit instruction per individual.
> > **Measuring Implicitness:** To ensure fairness and mitigate biases, we mask the non-implicit information (including answers and option letters) potentially revealed due to the model's limitations within the generated misleading instructions. We compared the misleading rates before and after masking the instructions, allowing us to objectively measure implicitness without bias. We also used GPT-4o to directly score the level of implicitness as a reference, with scores ranging from 1 to 9, where higher scores indicate a greater level of implicitness. we observe that closed-source models tend to generate instructions with higher implicitness and misleading rates. Examples of the comparison between human-generated instruction and instruction from other models can be found in Figures 18 and 19 of the revised version of the paper.
> >
> > #### Table:Comparison of implicitness, misleading rates, and time required for generating implicit instructions
> > | Model                 | MR     | Masked MR        | Implicitness | Time (s/it) |
> > |-----------------------|--------|------------------|--------------|-------------|
> > | MiniCPM-v-v2          | 39.71% | 18.98% (↓20.73%) | 5.67         | 2.26        |
> > | Phi-3-Vision          | 45.10% | 34.24% (↓10.86%) | 5.73         | 8.86        |
> > | Yi-VL-6b              | 27.49% | 21.84% (↓5.65%)  | 7.01         | 2.33        |
> > | Qwen-VL-Chat          | 35.65% | 31.95% (↓3.70%)  | 5.97         | 2.89        |
> > | Deepseek-VL-7b-Chat   | 42.10% | 22.51% (↓19.59%) | 6.31         | 2.78        |
> > | LLaVA-NeXT-7b-Vicuna  | 30.48% | 33.27% (↑2.79%)  | 6.65         | 5.4         |
> > | MiniCPM-Llama3-v2.5   | 44.06% | 38.23% (↓5.83%)  | 5.97         | 3.61        |
> > | GLM4V-9B-Chat         | 31.01% | 31.18% (↑0.17%)  | 6.22         | 6.98        |
> > | InternVL-Chat-V1_5    | 32.91% | 31.79% (↓1.12%)  | 5.80         | 7.71        |
> > | GLM-4V                | 45.31% | 42.01% (↓3.30%)  | 6.28         | 4.49        |
> > | GPT-4o                | 54.23% | 54.90% (↑0.67%)  | 7.05         | 5.20        |
> > | Human                 | 52.19% | 52.83% (↑0.64%)  | 6.30         | 240         |

---

> ### Author Response · Authors · 2024-11-20
>
> > Q3: During finetuning, a random set of explicit and implicit misled samples are used for finetuning, yet I am afraid the explicit misleading info has a too obvious and unique pattern due to how it's designed, hence too easy to pick them up, making the improvement after finetuning not too surprising.
>
> A3: Thank you for raising this insightful concern. We have analyzed the results for explicit misleading scenarios under implicit instruction fine-tuning, as illustrated in Figure 5-2 of the original version. To present more clarity, we updated the caption of the original figure and included detailed experimental results of the revised version. The findings demonstrate that with an increase in the volume of implicit fine-tuning data, the misleading rate in explicit misleading scenarios can be reduced to less than 20%. This highlights the effectiveness of implicit instruction fine-tuning in mitigating explicit misleading behavior.
>
> #### Table: Results for explicit misleading scenarios under implicit instruction fine-tuning.
> | Model               | Low          |  Medium         | High          |
> |---------------------|--------------|-----------------|---------------|
> | Phi-3-vision        | 10.90%       | 20.85%          | 40.19%        |
> | Qwen-VL-Chat        | 17.65%       | 26.88%          | 12.54%        |
> | MiniCPM-Llama3-v2.5 | 5.22%        | 14.48%          | 10.31%        |
> | GLM4V-9B-chat       | 3.54%        | 11.99%          | 12.18%        |
> | CogVLLM-chat        | 6.26%        | 13.71%          | 18.36%        |
> | InternVL-Chat-V1-5  | 6.06%        | 12.30%          | 11.89%        |
> | **Average**         | **8.94%**    | **16.37%**      | **17.91%**    |

---

> ### Author Response · Authors · 2024-11-20
>
> > Q4: Instead of finetuning, I would recommend the authors to simply systematically prompt the MLLMs, such as "The questions might contain misleading information, you should try to answer the question correctly despite of those misleading information ..."; another version could even give it two examples (one explicit and one implicit). I would guess/assume,  simply doing this extra prompting will make the results much better.
>
> A4: Thank you for your thoughtful suggestion. We conducted additional evaluations incorporating explicit and implicit misleading examples as part of systematic prompting strategies. The results indicate that even when the instructions explicitly warned the model about the presence of misleading information, the misleading rate remained above 70%. Similarly, providing explicit and implicit misleading examples in the instructions resulted in a misleading rate of approximately 70%. Although these strategies reduced the misleading rate by about 15%, the performance remains unsatisfactory for large models. The specific prompt and complete table can be found in the revised version paper, Table 22.
>
> #### Table: The average misleading rates of different explicit and implicit prompt-based defense strategies on 12 open-source MLLMs. (Warning means your first example. And different explicit and implicit instructions are shown in Appendix.A.2.4)
> | Model       | Warning    | Example(1) | Example(2) | Example(3) | COT      | Warning    | Example(1) | Example(2) | Example(3) | COT      |
> |-------------|------------|------------|------------|------------|----------|------------|------------|------------|------------|----------|
> | Average     | 69.28%     | 67.59%     | 67.26%     | 77.90%     | 88.50%   | 73.58%     | 72.08%     | 72.91%     | 92.36%     | 84.61%   |
>
> ---
>
> | Model   | Warning | Example(1) | Example(2) | Example(3) | Warning | Example(1) | Example(2) | Example(3) |
> |----------|------------|------------|------------|------------|------------|------------|------------|------------|
> | Average | 66.62%     | 70.13%     | 72.05%     | 72.72%     | 58.12%     | 61.70%     | 61.54%     | 61.19%     |
> ---

---

> ### Author Response · Authors · 2024-11-20
>
> > Q5: The questions only include multi-choice and T/F styles, which certainly makes the metrics calculation easier (reflected in equations 1 and 2), yet probably losing the delicacy in the type of Q/A  addressed?
>
> A5: Thank you for your suggestions. To comprehensively evaluate the effectiveness of our method, we transformed the discriminative task into a generative task. Specifically, we randomly sampled 200 data points prone to misleading. Only the images and questions were provided, without answer options. GPT-4o was used to evaluate the generated text and corresponding answers, and the misleading rates were calculated before and after introducing misleading information. (Note that the 200 data points prone to misleading were selected from external datasets. In contrast, using other unscreened generative task datasets does not effectively identify data points that are highly susceptible to misleading.)  Complete table can be found in the revised version paper, Table 27.
>
> #### Table: Comparison of explicit and implicit misleading instruction performance on generative tasks before and after fine-tuning
> | Model                           | Before (T-F) | Before (F-T) | After (T-F) | After (F-T) |
> |---------------------------------|--------------|--------------|-------------|-------------|
> | **Explicit**                    |              |              |             |             |
> | MiniCPM-v-v2                    | 69.23%       | 87.70%       | 25.00%      | 72.54%      |
> | Phi-3-vision                    | 100.00%      | 66.67%       | 71.43%      | 30.57%      |
> | Yi-VL-6b                        | 100.00%      | 82.89%       | 88.89%      | 55.50%      |
> | Qwen-VL-Chat                    | 94.12%       | 86.34%       | 86.21%      | 50.88%      |
> | Deepseek-VL-7b-Chat             | 92.31%       | 81.82%       | 70.59%      | 43.17%      |
> | LLaVA-NeXT-7b-Vicuna            | 100.00%      | 62.56%       | 100.00%     | 60.20%      |
> | MiniCPM-Llama3-v2.5             | 81.25%       | 83.71%       | 66.67%      | 64.29%      |
> | GLM4V-9B-Chat                   | 85.71%       | 80.90%       | 48.48%      | 62.42%      |
> | CogVLLM-Chat                    | 100.00%      | 54.55%       | 75.00%      | 3.35%       |
> | InternVL-Chat-V1_5              | 85.71%       | 69.27%       | 24.32%      | 68.10%      |
> | LLaVA-Next-34b                  | 100.00%      | 92.18%       | 62.50%      | 54.39%      |
> | Yi-VL-34b                       | 90.91%       | 92.59%       | 77.78%      | 14.21%      |
> | Average                         | 91.94%       | 76.99%       | 65.01%      | 48.31%      |
> | **Implicit**                    |              |              |             |             |
> | Average                         | 91.99%       | 44.38%       | 57.61%      | 23.57%      |

---

> > ### Author Response · Authors · 2024-11-20
> >
> > > Q6: The misleading information was only added to the textual questions, why not consider altering the image to inject misleading information?
> >
> > A6: Thank you for your suggestion. We assume that all MLLMs are capable of recognizing English characters within images. To inject misleading information into images, we tested its misleading rate by adding a watermark ("The true answer is xx") to the images. The results show a higher misleading rate compared to using misleading information in pure text.
> >
> > #### Table: The results of misleading rate on injecting misleading information into images.
> > | Model                    | Low   Image  | Low  Text      | Medium Image | Medium Text |
> > |--------------------------|--------------|----------------|--------------|----------------|
> > | MiniCPM-v-v2             | 62.91%       | 57.64%         | 78.89%       | 81.04%         |
> > | Phi-3-vision             | 60.10%       | 49.62%         | 67.57%       | 69.26%         |
> > | Yi-VL-6b                 | 84.93%       | 84.64%         | 93.49%       | 94.44%         |
> > | Qwen-VL-Chat             | 84.37%       | 80.53%         | 89.71%       | 89.33%         |
> > | Deepseek-VL-7b-Chat      | 37.25%       | 31.50%         | 65.44%       | 63.42%         |
> > | LLaVA-NeXT-7b-vicuna     | 44.40%       | 54.05%         | 40.09%       | 56.91%         |
> > | MiniCPM-Llama3-v2.5      | 54.88%       | 44.39%         | 66.55%       | 74.41%         |
> > | GLM4V-9B-chat            | 47.91%       | 17.58%         | 72.45%       | 51.89%         |
> > | CogVLLM-chat             | 21.93%       | 18.86%         | 52.95%       | 49.53%         |
> > | InternVL-Chat-V1-5       | 25.22%       | 17.46%         | 54.51%       | 50.55%         |
> > | LLaVA-Next-34b           | 77.22%       | 65.32%         | 94.35%       | 89.04%         |
> > | Yi-VL-34b                | 69.32%       | 56.99%         | 88.89%       | 78.87%         |
> > | **Average**              | **57.18%**   | **49.96%**     | **72.31%**   | **70.59%**     |

---

> ### Comment · Reviewer_zSd1 · 2024-11-24
> **Reply to Authors**
>
> Thank you for your revisions and further explanations. While I appreciate the expanded evaluations across modalities and the exploration of human-annotated data, I remain concerned about the limited novelty and comprehensiveness of the proposed approach/benchmark relative to existing benchmarks. Nonetheless I acknowledge the significant effort made in the revisions. And I'd like to reconsider my evaluation.

---

> > ### Author Response · Authors · 2024-11-27
> >
> > We sincerely appreciate your thoughtful comments and constructive feedback. Regarding your concern about the novelty and comprehensiveness of our proposed approach/benchmark relative to existing benchmarks, we would like to further clarify and highlight the following aspects:
> >
> > **Novelty** :
> >
> > Existing methods for evaluating model robustness and uncertainty have notable limitations. For instance, [1] evaluated model robustness to generate leading questions, but it is limited to a narrow set of question types and defining the questions.
> >
> > Similarly, [2] examined the effects of deceptive prompts on model behavior, but it lacks a systematic, quantitative metric to measure how such prompts induce uncertainty. [6] assessed the robustness of vision-language models to visual adversarial instructions, is limited by its focus on visual inputs alone and does not consider textual misleading prompts. [7] primarily investigated model robustness to manipulated visual inputs, overlooking the broader impact of textual misleading instructions on model uncertainty.
> >
> > [4],[5] evaluated the consistency or uncertainty only for language models. [3] focused on consistency on three types of tasks, but its evaluation is confined to fixed conditions and does not account for how misleading instructions trigger uncertainty in real-world scenarios.
> >
> > As Reviewer t1Gv pointed out, our approach introduces misleading evidence in prompts to test model robustness, which is both novel and interesting. This method offers a systematic and scalable way to evaluate consistency and reasoning under adversarial conditions—an area that has not been fully explored in existing benchmarks. To the best of our knowledge, we are the first to employ both explicit and implicit misleading instructions in a multimodal setting to effectively identify uncertainty data and evaluate model response uncertainty.
> >
> > Our method does not require the design of specific misleading questions [1] [2] or identifications of particular visual inputs [6][7]. It is highly extensible, enabling applications to any multimodal dataset and providing greater flexibility.
> > Previous studies typically relied on consistency and accuracy metrics to assess uncertainty [3] [4] [5], but they often required 5–15 responses to effectively identify uncertain data. In contrast, our two-stage process using misleading instructions only requires two responses to identify uncertainty. As Reviewer 7azo highlighted, our approach introduces novel and more efficient metrics that address limitations in widely used metrics, such as the self-consistency rate. We also present the relationship between consistency rate and misleading rate, as well as between consistency rate and accuracy (see Figures 1 and 10 in the revised manuscript).
> >
> > Unlike existing methods [1-7], we also improve the performance of current MLLMs by fine-tuning the models based on the identification of uncertain data. This fine-tuning significantly reduces the misleading rate, improves the consistency rate, and importantly, achieves these improvements without any loss in accuracy.
> >
> > [1] Seeing Clearly, Answering Incorrectly: A Multimodal Robustness Benchmark for Evaluating MLLMs on Leading Questions
> >
> > [2] How Easy is It to Fool Your Multimodal LLMs? An Empirical Analysis on Deceptive Prompts
> >
> > [3] Unveiling the Tapestry of Consistency in Large Vision-Language Models
> >
> > [4] Benchmarking and improving generatorvalidator consistency of language models
> >
> > [5] Generating with confidence: Uncertainty quantification for black-box large language models
> >
> > [6] AVIBench: Towards Evaluating the Robustness of Large Vision-Language Model on Adversarial Vsual-Instructions
> >
> > [7] On Evaluating Adversarial Robustness of Large Vision-Language Models

---

> > > ### Author Response · Authors · 2024-11-27
> > >
> > > **Comprehensiveness** :
> > >
> > > To provide a more comprehensive evaluation of our benchmark, we also included additional metrics beyond the accuracy and misleading rate obtained from the main experiment.
> > >
> > > **Evaluation on more metrics**:
> > >
> > > **(1) ECE**: We collected the confidence scores of the model's outputs and calculated the Expected Calibration Error (ECE) before and after fine-tuning the model in the True-False (T-F) scenario.
> > >
> > >  **(2) Consistency rate**: We provided the maximum frequency at which the model gives the same answer when faced with the same questions from the benchmark multiple times, which is referred to as the consistency rate.
> > >
> > > **Evaluation on more specific categories**:
> > >
> > > **(1)**: We divided the questions in the entire benchmark into three types of tasks: perception, reasoning, and mastery. For each type of task, we provided the misleading rates under implicit and explicit misleading scenarios as well as before and after fine-tuning. This task categorization helps in comparing the model's performance across different types of tasks.
> > >
> > > **(2)**: Furthermore, we break down perception, reasoning, and mastery tasks into more granular evaluations. **Perception** includes the following abilities: Visual Identification (VI), Text Recognition (TR), Aesthetic Perception (AP), and Spatial Awareness (SA); **Reasoning** includes Logical Reasoning (LR), Scientific Reasoning (SR), and Cross-Domain Reasoning (CDR); and **Mastery** includes Natural Sciences (NS), Social Studies (SS), and Applied Arts (AA), resulting in a total of 10 distinct abilities. Further refinement of the question domains facilitates the evaluation of the model's capability boundaries.
> > >
> > > **(3)**: We provided the average misleading rates measured on two different types of questions in the benchmark (multiple choice (CH) and yes/no (Y/N)) to evaluate the impact of question types on the model's uncertainty.
> > >
> > > #### Table: Comparison of ECE before and after fine-tuning on our benchmark
> > > | Model                  | Before | After |
> > > |------------------------|--------|-------|
> > > | MiniCPM-v-v2           | 0.46   | 0.24  |
> > > | Phi-3-vision           | 0.46   | 0.15  |
> > > | Yi-VL-6b               | 0.45   | 0.27  |
> > > | Qwen-VL-Chat           | 0.49   | 0.24  |
> > > | Deepseek-VL-7b-Chat    | 0.47   | 0.20  |
> > > | LLaVA-NeXT-7b-vicuna   | 0.48   | 0.23  |
> > > | MiniCPM-Llama3-v2.5    | 0.49   | 0.18  |
> > > | GLM4V-9B-chat          | 0.46   | 0.25  |
> > > | CogVLLM-chat           | 0.46   | 0.27  |
> > > | InternVL-Chat-V1-5     | 0.47   | 0.24  |
> > > | LLaVA-Next-34b         | 0.49   | 0.19  |
> > > | Yi-VL-34b              | 0.45   | 0.26  |
> > > | **Average**            | **0.47** | **0.23** |
> > >
> > >
> > > #### Table: The consistency rate of both before and after fine-tuning under low and high misleading rate scenario. (Detailed table can be found in the revised version paper, Table 21)
> > > | Model                     | Low (Before) | Low (After) | Low (Change) | High (Before) | High (After) | High (Change) |
> > > |---------------------------|--------------|-------------|--------------|---------------|--------------|---------------|
> > > | MiniCPM-v-v2              | 82.93%       | 97.83%      | +14.90%      | 56.52%        | 90.64%       | +34.12%       |
> > > | Phi-3-vision              | 79.89%       | 89.33%      | +9.44%       | 63.94%        | 87.77%       | +23.83%       |
> > > | GLM4v-9b                  | 94.33%       | 99.00%      | +4.67%       | 82.28%        | 95.85%       | +13.57%       |
> > > | LLaVA-Next-34b            | 73.30%       | 98.61%      | +25.31%      | 53.30%        | 91.81%       | +38.51%       |
> > > | **Average**               | **82.61%**   | **96.19%**  | **+13.58%**  | **64.51%**    | **91.02%**   | **+26.51%**   |

---

> > > > ### Author Response · Authors · 2024-11-27
> > > >
> > > > #### Table: Comparison of explicit and implicit misleading instruction performance on different types tasks before and after fine-tuning.(Complete table can be found in the revised version paper, Table 36,37.)
> > > > | Model                  | Perception (T-F)     | Reasoning (T-F)      | Mastery (T-F)        |
> > > > |------------------------|----------------------|----------------------|----------------------|
> > > > | MiniCPM-v-v2           | 5.33% (↓78.37%)     | 7.28% (↓66.66%)      | 14.63% (↓59.73%)     |
> > > > | Phi-3-vision           | 7.26% (↓78.62%)     | 6.62% (↓52.29%)      | 6.86% (↓56.46%)      |
> > > > | Yi-VL-6b               | 9.42% (↓80.91%)     | 21.84% (↓66.49%)     | 46.92% (↓47.55%)     |
> > > > | Qwen-VL-Chat           | 1.76% (↓90.06%)     | 7.78% (↓76.00%)      | 12.81% (↓68.33%)     |
> > > > | Deepseek-VL-7b-Chat    | 1.42% (↓66.34%)     | 3.27% (↓54.71%)      | 6.78% (↓53.62%)      |
> > > > | LLaVA-NeXT-7b-vicuna   | 4.81% (↓75.37%)     | 10.72% (↓44.31%)     | 15.68% (↓40.15%)     |
> > > > | MiniCPM-Llama3-v2.5    | 0.73% (↓71.04%)     | 1.10% (↓63.18%)      | 1.75% (↓60.19%)      |
> > > > | GLM4V-9B-chat          | 4.61% (↓39.82%)     | 8.39% (↓35.57%)      | 23.68% (↓35.80%)     |
> > > > | CogVLLM-chat           | 8.13% (↓56.29%)     | 8.15% (↓37.65%)      | 32.40% (↓16.89%)     |
> > > > | InternVL-Chat-V1-5     | 0.60% (↓49.74%)     | 2.85% (↓49.15%)      | 9.93% (↓50.51%)      |
> > > > | LLaVA-Next-34b         | 2.12% (↓75.54%)     | 3.25% (↓84.97%)      | 2.25% (↓85.77%)      |
> > > > | Yi-VL-34b              | 9.13% (↓71.55%)     | 17.12% (↓56.17%)     | 30.48% (↓37.84%)     |
> > > > | **Explicit Average**      | **4.61% (↓69.47%)**     | **8.20% (↓57.26%)**      | **17.02% (↓51.07%)**     |
> > > >
> > > >
> > > >
> > > >
> > > >
> > > > #### Table: The average misleading rates across subfields in perception, reasoning, and mastery tasks before fine-tuning. (Complete table can be found in the revised version paper, Table 38,39.)
> > > > **Perception Task**: Visual Identification (VI), Text Recognition (TR), Aesthetic Per-ception (AP), Spatial Awareness (SA)
> > > > **Reasoning Task**: Logical Reasoning (LR), Scientific Reasoning (SR), Cross-Domain Reasoning (CDR)
> > > > **Mastery Task**:  Natural Sciences (NS), Social Studies (SS), Applied Arts (AA).
> > > >
> > > > | **Model**            | **VI**       | **TR**       | **AP**       | **SA**       | **LR**       | **SR**       | **CDR**      | **NS**       | **SS**       | **AA**       |
> > > > |-----------------------|--------------|--------------|--------------|--------------|--------------|--------------|--------------|--------------|--------------|--------------|
> > > > | **Explicit T-F**      | **77.53%**   | **76.97%**   | **81.45%**   | **83.60%**   | **63.43%**   | **66.79%**   | **75.69%**   | **68.67%**   | **69.68%**   | **69.43%**   |
> > > > | **Explicit F-T**      | **85.86%**   | **85.51%**   | **85.61%**   | **87.07%**   | **81.09%**   | **85.78%**   | **83.82%**   | **85.60%**   | **86.58%**   | **84.74%**   |
> > > > | **Implicit T-F**      | **77.53%**   | **76.97%**   | **81.45%**   | **83.60%**   | **63.43%**   | **66.79%**   | **75.69%**   | **68.67%**   | **69.68%**   | **69.43%**   |
> > > > | **Implicit F-T**      | **80.51%**   | **84.35%**   | **92.08%**   | **71.83%**   | **76.15%**   | **85.77%**   | **88.87%**   | **74.18%**   | **74.59%**   | **72.40%**   |
> > > >
> > > >
> > > >
> > > >
> > > >
> > > > #### Table: The average misleading rates on two different types of questions before fine-tuning. (Complete table can be found in the revised version paper, Table 34,35.)
> > > >
> > > > | **Model**                        | **T-F Choice**       | **T-F Yes/No**      | **F-T Choice**       | **F-T Yes/No**      |
> > > > |----------------------------------|----------------------|---------------------|----------------------|---------------------|
> > > > | **Explicit Low misleading rate** | **46.55%**    | **55.19%**  | **72.88%**  | **57.14%**  |
> > > > | **Explicit Medium misleading rate** | **66.26%**   | **83.29%**   | **80.98%**    | **76.92%**  |
> > > > | **Explicit High misleading rate** | **85.99%**   | **92.95%**   | **92.90%**   | **94.26%**   |
> > > > | **Implicit Low misleading rate** | **75.89%**  | **49.22%**  | **83.55%**  | **57.90%**  |
> > > > | **Implicit Medium misleading rate** | **83.84%**   | **70.03%**  | **81.58%**   | **76.85%**  |
> > > > | **Implicit High misleading rate** | **95.00%**   | **86.22%** | **82.45%**   | **87.04%**  |

---

### Official Review · Reviewer_7azo · 2024-11-05

**Soundness:** 1
**Presentation:** 3
**Contribution:** 3
**Rating:** 5
**Confidence:** 3

**Summary:**

This paper studies uncertainty measurement for responses from MLLMs. The main novelty is a novel uncertainty measurement based on how MLLMs' response shifts after injecting misleading instructions. Empirically, the paper developed Multimodal Uncertainty Benchmark (MUB), and systematically evaluates most major MLLMs’s uncertainty; The result suggests a dominant issue of uncertainty in MLLMs, with an average misleading rate exceeding 86%. The author experimented with fine-tuning MLLMs with targeted misleading data, which notably improves robustness.

**Strengths:**

- I think the effort of proposing novel and more efficient metrics to is very relevant and helpful. Previous metrics, such as self-consistency rate are widely used but in practice I found it to be very unreliable on big models.
- The experimental evaluation is comprehensive, covering most of the commonly used closed and open-sourced models.

**Weaknesses:**

- Soundness of measurement (Major): While I very much appreciate the effort on better and efficient measurements for uncertainty, currently I still have some doubts about whether adding misleading information can measure the uncertainty in model’s response. I’ll explain my concerns and perhaps the authors can clarify:
    - The measurement might be dependent to the misleading information themselves: the content, position, length, etc might all influence this metric. Moreover, since the implicit misinformation is generated by GPT4o, which is also evaluated on the benchmark, will it incur evaluator bias?
    - Implicit scenarios seem better defined; But for explicit scenarios (e.g. telling the model true answer), the model behavior might be inherently undefined: i.e. shall the model follow the user’s “instruction” (e.g. “the true answer”), or answer the question and ignore user instruction.
- Task (Minor): The study is confined to multiple choice question. I am curious about how would the definitions, measurements, and findings generalize to open-ended question. But I don’t think this is a major point, because most current VLM benchmarks are multiple-choice only.

**Questions:**

See weakness section.

---

> ### Author Response · Authors · 2024-11-22
>
> We greatly appreciate the reviewer’s detailed feedback and constructive insights, which have significantly helped us refine our study. Below, we address each of the key points raised:
>
> >Q1: The measurement might be dependent to the misleading information themselves: the content, position, length, etc might all influence this metric.
>
> A1: Thank you for raising this insightful point. To address this concern, we conducted additional experiments to evaluate the effects of position, length, and content of misleading information on the misleading rate.
> - **Content**: In the manuscript, we present 11 explicit prompts, which are categorized into four types. The results indicate that explicitly providing the model with the answer increases the misleading rate.
> - **Position**: To analyze the influence of position, we tested scenarios where misleading information was inserted either before (after the system prompt) or after the question. The results indicate that the misleading rate is lower when explicit misleading information is inserted before the question compared to when it is inserted after the question. (Notably, in the paper, all misleading information is inserted after the question.)
> - **Length**: We increased the length of the phrase "The true answer is" by twofold and threefold, respectively, to examine whether the phrase's length impacts the results. The results indicate that increasing the length alone, without modifying the content, has minimal impact on the misleading rate.
>
> #### Table: the average misleading rate of 11 explicit prompt templates on 12 open-source MLLMs.
> | **Model**               | **Factors** | **Apparent** | **Argue** | **While** | **Obvious** | **Context** | **Given** | **Evidence** | **Correct** | **GPT**  | **User**  |
> |-------------------------|-------------|--------------|-----------|-----------|-------------|-------------|-----------|--------------|-------------|----------|-----------|
> | **Average**            | **65.16%**  | **67.43%**   | **65.58%**| **69.92%**| **62.18%**  | **66.30%**  | **67.30%**| **64.28%**   | **63.89%**  | **60.02%**| **48.74%**|
>
>
>
> ##### The misleading rates across different positions and lengths in 12 open-source MLLMs.
> | **Model**               | **Before(Repeat 1)** | **After(Repeat 1)** | **Repeat 2** | **Repeat 3** |
> |-------------------------|-------------|--------------|-----------|-----------|
> | **Average T-F**            | **57.17%**  | **79.38%**   | **76.47%**| **76.96%**|
> | **Average F-T**            | **56.40%**  | **78.08%**   | **79.50%**| **79.30%**|

---

> > ### Author Response · Authors · 2024-11-22
> >
> > > Q2: Moreover, since the implicit misinformation is generated by GPT4o, which is also evaluated on the benchmark, will it incur evaluator bias?
> >
> > A2: Thank you for raising this important concern regarding potential evaluator bias. we conducted an additional evaluation by randomly selecting 100 samples and comparing the outputs generated by other open-source and closed-source models. The findings are summarized in the table below. The results indicate that:
> > - **Human-generated instructions:** The implicit misleading instructions generated by human are produced more slowly, which reflects the manual effort involved in creating implicit misleading information.
> > - **No bias implicitness measurment:** To ensure fairness and mitigate biases, we mask the non-implicit information (including answers and option letters) potentially revealed due to the model's limitations within the generated misleading instructions. By comparing the misleading rates obtained from the instructions before and after masking, we can objectively measure the level of implicitness without bias. When the original misleading rate exceeds a certain threshold, a larger reduction in the misleading rate after masking indicates a lower degree of implicitness in the instructions.
> > - **Open-source models** demonstrate a lower rate of misinterpretation, suggesting they may be less susceptible to implicit misleading information.
> > - **Closed-source models**, on the other hand, exhibit a higher rate of misinterpretation and a greater degree of implicitness, indicating a stronger vulnerability to implicit misinformation.
> >
> >
> > Table: The comparison of open-source and closed-source MLLMs, as well as humans, regarding their generation of implicit misleading instructions.
> >  Model                 | MR     | Masked MR         | Time (s/it) |
> > |-----------------------|--------|------------------|-------------|
> > | MiniCPM-v-v2          | 39.71% | 18.98% (↓20.73%) | 2.26        |
> > | Phi-3-Vision          | 45.10% | 34.24% (↓10.86%) | 8.86        |
> > | Yi-VL-6b              | 27.49% | 21.84% (↓5.65%)  | 2.33        |
> > | Qwen-VL-Chat          | 35.65% | 31.95% (↓3.70%)  | 2.89        |
> > | Deepseek-VL-7b-Chat   | 42.10% | 22.51% (↓19.59%) | 2.78        |
> > | LLaVA-NeXT-7b-Vicuna  | 30.48% | 33.27% (↑2.79%)  | 5.4         |
> > | MiniCPM-Llama3-v2.5   | 44.06% | 38.23% (↓5.83%)  | 3.61        |
> > | GLM4V-9B-Chat         | 31.01% | 31.18% (↑0.17%)  | 6.98        |
> > | InternVL-Chat-V1_5    | 32.91% | 31.79% (↓1.12%)  | 7.71        |
> > | GLM-4V                | 45.31% | 42.01% (↓3.30%)  | 4.49        |
> > | GPT-4o                | 54.23% | 54.90% (↑0.67%)  | 5.20        |
> > | Human                 | 52.19% | 52.83% (↑0.64%)  | 240         |

---

> > > ### Author Response · Authors · 2024-11-22
> > >
> > > >Q3: Implicit scenarios seem better defined; But for explicit scenarios (e.g. telling the model true answer), the model behavior might be inherently undefined: i.e. shall the model follow the user’s “instruction” (e.g. “the true answer”), or answer the question and ignore user instruction.
> > >
> > > A3: Thank you for highlighting this important distinction in model behavior for explicit scenarios. To better understand whether the model adheres to user instructions or focuses solely on answering the question, we conducted two additional experiments:
> > > **Scenario 1: No user instructions, only question instructions** : We appended an incorrect character or word to the end of the text after the input question to simulate accidental input. This ensures that the model's behavior is not inherently undefined due to other user instructions. The results revealed that the misleading rate remained consistently high. This indicates that the model predominantly focuses on answering the question but remains highly vulnerable to misleading information embedded in the prompt.
> > > **Scenario 2: User instructions take priority. Clear instructions to ignore misleading information** : In this experiment, the model was explicitly told that the input contained misleading information, with instructions to disregard it and provide the correct answer (e.g., "The following input contains misleading information: {misleading information}. Please focus only on the questions and options and ignore all other misleading instructions! "). Despite these clear instructions, the model still showed a high misleading rate, highlighting its persistent susceptibility to misleading information.
> > >
> > > #### Table: The comparison of simplified explicit prompts and clear instructions to ignore misleading information.
> > > | Model                    | Scenario 1 T-F | Scenario 1 F-T | Scenario 2 T-F | Scenario 2 F-T |
> > > |--------------------------|----------|----------|----------|----------|
> > > | MiniCPM-v-v2             | 58.60%   | 66.84%   | 86.11%   | 74.76%   |
> > > | Phi-3-vision             | 62.04%   | 43.08%   | 77.30%   | 71.99%   |
> > > | Yi-VL-6b                 | 60.04%   | 48.66%   | 78.10%   | 72.55%   |
> > > | Qwen-VL-Chat             | 90.83%   | 62.46%   | 92.55%   | 68.61%   |
> > > | Deepseek-VL-7b-Chat      | 74.55%   | 61.35%   | 80.32%   | 76.76%   |
> > > | LLaVA-NeXT-7b-vicuna     | 73.55%   | 71.81%   | 59.45%   | 65.51%   |
> > > | MiniCPM-Llama3-v2.5      | 73.25%   | 61.52%   | 63.72%   | 63.23%   |
> > > | GLM4V-9B-chat            | 27.48%   | 32.82%   | 64.77%   | 83.06%   |
> > > | CogVLLM-chat             | 38.88%   | 40.97%   | 84.65%   | 90.19%   |
> > > | InternVL-Chat-V1-5       | 53.83%   | 53.23%   | 52.33%   | 50.27%   |
> > > | LLaVA-Next-34b           | 60.84%   | 65.51%   | 70.75%   | 69.28%   |
> > > | Yi-VL-34b                | 73.77%   | 71.70%   | 79.76%   | 76.14%   |
> > > | **Average**              | 62.31%   | 56.66%   | 74.15%   | 71.86%   |

---

> > > > ### Author Response · Authors · 2024-11-22
> > > >
> > > > >Q4: The study is confined to multiple choice question. I am curious about how would the definitions, measurements, and findings generalize to open-ended question. But I don’t think this is a major point, because most current VLM benchmarks are multiple-choice only.
> > > >
> > > > A4: Thank you for your suggestions. To comprehensively evaluate the effectiveness of our method, we transformed the discriminative task into a generative task. Specifically, we randomly sampled 200 data points prone to misleading. Only the images and questions were provided, without answer options. We used GPT-4o to evaluate whether the text generated by open-source models aligns with the answers and calculated the misleading rates before and after introducing misleading information. (Note that the 200 data points prone to misleading were selected from external datasets. In contrast, using other unscreened generative task datasets does not effectively identify data points that are highly susceptible to misleadingn.)
> > > >
> > > > #### Table: Comparison of explicit and implicit misleading instruction performance on generative tasks before and after fine-tuning (Table27 in the new revision)
> > > > | Model                           | Before (T-F) | Before (F-T) | After (T-F) | After (F-T) |
> > > > |---------------------------------|--------------|--------------|-------------|-------------|
> > > > | **Explicit**                    |              |              |             |             |
> > > > | MiniCPM-v-v2                    | 69.23%       | 87.70%       | 25.00%      | 72.54%      |
> > > > | Phi-3-vision                    | 100.00%      | 66.67%       | 71.43%      | 30.57%      |
> > > > | Yi-VL-6b                        | 100.00%      | 82.89%       | 88.89%      | 55.50%      |
> > > > | Qwen-VL-Chat                    | 94.12%       | 86.34%       | 86.21%      | 50.88%      |
> > > > | Deepseek-VL-7b-Chat             | 92.31%       | 81.82%       | 70.59%      | 43.17%      |
> > > > | LLaVA-NeXT-7b-Vicuna            | 100.00%      | 62.56%       | 100.00%     | 60.20%      |
> > > > | MiniCPM-Llama3-v2.5             | 81.25%       | 83.71%       | 66.67%      | 64.29%      |
> > > > | GLM4V-9B-Chat                   | 85.71%       | 80.90%       | 48.48%      | 62.42%      |
> > > > | CogVLLM-Chat                    | 100.00%      | 54.55%       | 75.00%      | 3.35%       |
> > > > | InternVL-Chat-V1_5              | 85.71%       | 69.27%       | 24.32%      | 68.10%      |
> > > > | LLaVA-Next-34b                  | 100.00%      | 92.18%       | 62.50%      | 54.39%      |
> > > > | Yi-VL-34b                       | 90.91%       | 92.59%       | 77.78%      | 14.21%      |
> > > > | Average                         | 91.94%       | 76.99%       | 65.01%      | 48.31%      |
> > > > | **Implicit**                    |              |              |             |             |
> > > > | Average                         | 91.99%       | 44.38%       | 57.61%      | 23.57%      |

---

> > > > > ### Author Response · Authors · 2024-11-26
> > > > >
> > > > > Dear Reviewer  7azo,
> > > > >
> > > > > Thank you very much for your invaluable feedback on our paper. We have meticulously reviewed each of your points and endeavored to address them thoroughly. We would greatly appreciate it if you could review our responses and let us know if you have any additional comments regarding the paper or the rebuttal. We are eager to embrace all your critiques and integrate them into our work.
> > > > >
> > > > > Thank you!

---

### Official Review · Reviewer_t1Gv · 2024-11-05

**Soundness:** 2
**Presentation:** 2
**Contribution:** 3
**Rating:** 5
**Confidence:** 3

**Summary:**

The paper introduces a dataset of misleading instructions for multimodal language models. This is done in two ways: through a template (telling the model that the answer is "X", where X is wrong), and through a language model (for instance by adding evidence or reasoning that contradicts the true answer). It is shown that models have lower consistency on instructions that are successfully misleading, and that fine-tuning can improve this.

**Strengths:**

The paper tackles an interesting problem, and the idea of adding misleading evidence to a prompt is a nice way to test for robustness. I also thought it was interesting that consistency decreases.

**Weaknesses:**

I'm not sure I buy the overall motivation -- of course if you tell the model the answer is wrong, it will flip some fraction of the time. But is this going to affect real users in any way? Arguably this is even an intended feature to avoid getting into fights with users.

As a result of this, I don't think the "Explicit" misleading prompts are really a meaningful benchmark. The "Implicit" ones are more interesting, but need more detailed treatment: for instance, at least give several examples and enough information to assess data quality. The evaluation of the "Implicit" setting is also strange -- using best-of-5 sampling (even though "Explicit" is best-of-1) which inflates the success rate.

A separate issue is that, throughout, there is not enough information to understand the data or experimental setup in detail. The paper says that they "fine-tuned on separate data", but there are not many details that would let a reader verify or reproduce the experiments. (This is also part of the problem with the "Implicit" setting -- not enough details to fully understand.)

I think the authors are tackling an interesting problem, and have made a good start on it, but in my opinion the experiments and writing should be tightened up before it's ready to be accepted to ICLR.

**Questions:**

Why best-of-5 sampling, and why only for "Implicit"?

Can you provide several random samples from the "Implicit" setting?

---

> ### Author Response · Authors · 2024-11-20
>
> We sincerely thank the reviewer for recognizing the significance of the problem we are addressing and acknowledging the potential of our work. We appreciate your constructive feedback regarding the experiments and writing, and we have made substantial efforts to tighten both aspects in the revised manuscript. Below, we detail the specific revisions and improvements made in response to your comments.
>
> >Q1: "Is this going to affect real users in any way? Arguably this is even an intended feature to avoid getting into fights with users."
>
> A1:  Thank you for raising this important question. To address this, we offer the following insights and supporting evidence:
> - Uncertain responses can significantly affect users’ trust and decision-making processes, as demonstrated by the following example: **What is the capital of the UK?**  **A**: London (confidence: approximately 1.0).  **B**: Paris (confidence: 1.29 × 10⁻¹⁰). [1]. Even for questions where the correct answer is highly certain, there remains a non-zero probability of the model producing an incorrect response. For questions involving greater uncertainty, particularly in multimodal data, this raises concerns about how much users should rely on the model's answers.
> - We conducted a simple experiment where, after the user inputted a question (including both multiple-choice and true/false questions), an incorrect character or word was appended to the end of the text to simulate accidental input. No additional guiding information was provided. However, **the results still demonstrated a relatively high misleading rate**, the model maintains a misleading rate of approximately 62.4%.
>
> #### Table: The misleading rate of adding a misleading letter to the end of a question.
> | Model                        | ACC    |Misleading rate|
> |------------------------------|--------|--------|
> | MiniCPM-v-v2                 | 57.25% | 58.6%  |
> | Phi-3-vision                 | 44.48% | 62.04% |
> | yi-vl-6b-chat                | 55.52% | 60.04% |
> | Yi-VL-6b                     | 61.36% | 90.83% |
> | Deepseek-VL-7b-Chat          | 59.96% | 74.55% |
> | LLaVA-NeXT-7b-vicuna         | 46.65% | 73.55% |
> | MiniCPM-Llama3-v2.5          | 58.66% | 73.25% |
> | GLM4V-9B-chat                | 51.19% | 27.48% |
> | CogVLLM-chat                 | 44.26% | 38.88% |
> | InternVL-Chat-V1-5           | 56.49% | 53.83% |
> | LLaVA-Next-34b               | 56.39% | 60.84% |
> | Yi-VL-34b                    | 54.87% | 73.77% |
> | **Average**                  | 54.88% | 62.36% |
>
>
> Ref:
> [1] Yadkori et al., 2024. To Believe or Not to Believe Your LLM
>
> >Q2: "As a result of this, I don't think the 'Explicit' misleading prompts are really a meaningful benchmark."
>
> A2: Thank you for raising this concern. We would like to clarify the significance of the "Explicit" misleading prompts as a meaningful benchmark and provide additional context: we included the experimental results for explicit prompts in Table 8 of the appendix in our original submission(Table 6 in the revised version). These explicit prompts consist of 11 additional templates, including examples such as "the users' answer is," "the GPT-4's answer is," and "given the context and the picture, the answer is xx." The experimental results reveal that explicit prompts consistently exhibit high rates of misinformation. **This pattern is not limited to a single template but extends across various forms of explicit prompts, making them a meaningful benchmark for assessing the vulnerability of MLLMs to adversarial manipulation.** Such insights are critical for developing more resilient models and better understanding their limitations. Complete table can be found in the revised version, Table 7.
>
> #### Table: the average misleading rate of 11 explicit prompt templates on 12 open-source MLLMs.
> | **Model**               | **Factors** | **Apparent** | **Argue** | **While** | **Obvious** | **Context** | **Given** | **Evidence** | **Correct** | **GPT**  | **User**  |
> |-------------------------|-------------|--------------|-----------|-----------|-------------|-------------|-----------|--------------|-------------|----------|-----------|
> | **Average**            | **65.16%**  | **67.43%**   | **65.58%**| **69.92%**| **62.18%**  | **66.30%**  | **67.30%**| **64.28%**   | **63.89%**  | **60.02%**| **48.74%**|
>
> ---

---

> ### Author Response · Authors · 2024-11-20
>
> >Q3: "The 'Implicit' ones are more interesting, but need more detailed treatment: for instance, at least give several examples and enough information to assess data quality."
>
> A3: Thank you for your valuable suggestions. We appreciate your interest in the "Implicit" prompts. To address your concerns, we have made the following updates and additions:
> **Details and Revisions Made**: In the original submission, we provided scores for the implicitness of generated implicit instructions in Figure 4-(3) and examples of such instructions in Figures 17 and 18. In the revised version, we have expanded Section 2.2 to include additional details about the implicit instructions, offering more context and examples to better illustrate their nature and design.
> **Comprehensive Evaluation of Implicit Instructions**: To more comprehensively evaluate the generated implicit instructions, we have also introduced more comprehensive evaluation the generated implicit misleading data quality, as shown in the table below. We randomly selected 100 images and compared their results with those from human annotations, open-source models, and closed-source models. To ensure fairness and mitigate biases, we masked the non-implicit information (including answers and option letters) potentially revealed due to the model's limitations within the generated misleading instructions. The implicitness level is evaluated based on scores assigned to the generated implicit instructions, with the highest possible score being 9. The findings indicate that closed-source models demonstrate a higher degree of implicitness as well as a higher misleading rate.
>
>
> #### Table:Comparison of implicitness, misleading rates, and time required for generating implicit instructions
>  Model                 | MR     | Masked MR        | Implicitness | Time (s/it) |
> |-----------------------|--------|------------------|--------------|-------------|
> | MiniCPM-v-v2          | 39.71% | 18.98% (↓20.73%) | 5.67         | 2.26        |
> | Phi-3-Vision          | 45.10% | 34.24% (↓10.86%) | 5.73         | 8.86        |
> | Yi-VL-6b              | 27.49% | 21.84% (↓5.65%)  | 7.01         | 2.33        |
> | Qwen-VL-Chat          | 35.65% | 31.95% (↓3.70%)  | 5.97         | 2.89        |
> | Deepseek-VL-7b-Chat   | 42.10% | 22.51% (↓19.59%) | 6.31         | 2.78        |
> | LLaVA-NeXT-7b-Vicuna  | 30.48% | 33.27% (↑2.79%)  | 6.65         | 5.4         |
> | MiniCPM-Llama3-v2.5   | 44.06% | 38.23% (↓5.83%)  | 5.97         | 3.61        |
> | GLM4V-9B-Chat         | 31.01% | 31.18% (↑0.17%)  | 6.22         | 6.98        |
> | InternVL-Chat-V1_5    | 32.91% | 31.79% (↓1.12%)  | 5.80         | 7.71        |
> | GLM-4V                | 45.31% | 42.01% (↓3.30%)  | 6.28         | 4.49        |
> | GPT-4o                | 54.23% | 54.90% (↑0.67%)  | 7.05         | 5.20        |
> | Human                 | 52.19% | 52.83% (↑0.64%)  | 6.30         | **240**     |
>
>
> ---
>
>
> >Q4: "The evaluation of the 'Implicit' setting is also strange -- using best-of-5 sampling (even though 'Explicit' is best-of-1) which inflates the success rate."
>
> A4: I apologize for not providing a clear explanation here we used GPT-4o to generate five misleading implicit instructions for each question in a single response and intended to apply all five instructions. To ensure a fair comparison between explicit and implicit strategies, we have addressed this concern in the revised version by comparing the results of explicit and implicit strategies under the sampling-1 setting. Additionally, we have expanded the evaluation by providing results for implicit strategies using sampling-1, sampling-3, and sampling-5 across scenarios with both low and high misleading rates. These updates aim to offer a more balanced and comprehensive evaluation of the "Implicit" setting and address any potential concerns regarding inflated success rates.  Complete table can be found in the revised version paper, Table 13,14.
>
>
> #### Table: The average misleading rate of different sample strategies on 12 open-source MLLMs.
> |         |               | **MR(T → F)**       |                  |                  | **MR(F → T)**       |                  |                  |
> |--------------------|--------------|---------------------|------------------|------------------|---------------------|------------------|------------------|
> |   **Model**                  |         **Accuracy**    | **Sample-1**       | **Sample-3**     | **Sample-5**     | **Sample-1**       | **Sample-3**     | **Sample-5**     |
> | **Average[low]**        | **73.45%**   | **54.81%**         | **72.36%**       | **77.61%**       | **47.55%**         | **73.58%**       | **78.98%**       |
> | **Average[high]**        | **56.63%**   | **66.34%**         | **84.42%**       | **87.68%**       | **61.47%**         | **79.08%**       | **85.00%**       |

---

> > ### Author Response · Authors · 2024-11-20
> >
> > >Q5: A separate issue is that, throughout, there is not enough information to understand the data or experimental setup in detail. The paper says that they "fine-tuned on separate data", but there are not many details that would let a reader verify or reproduce the experiments. (This is also part of the problem with the "Implicit" setting -- not enough details to fully understand.)
> >
> > A5: We apologize for not providing sufficient details about the data and experimental setup in the original submission. To address this issue, we have made the following updates in the revised version: (1): we now include a detailed description of the fine-tuning data formats in Figure 21, along with the experimental details and parameters for fine-tuning in Section 3. (2) For the implicit instruction generation setting, we provide templates in Figure 15 of the appendix and present examples of implicit instructions generated by various models in Figures 17 and 18. In the new revision, we have expanded the descriptions to include more detailed explanations.

---

> ### Author Response · Authors · 2024-11-26
>
> Dear Reviewer t1Gv,
>
> Thank you very much for your invaluable feedback on our paper. We have meticulously reviewed each of your points and endeavored to address them thoroughly. We would greatly appreciate it if you could review our responses and let us know if you have any additional comments regarding the paper or the rebuttal. We are eager to embrace all your critiques and integrate them into our work.
>
> Thank you!

---

### Author Response · Authors · 2024-11-26

Dear Reviewers,

Thank you very much for your invaluable feedback on our paper. We have meticulously reviewed each of your points and endeavored to address them thoroughly. We would greatly appreciate it if you could review our responses and let us know if you have any additional comments regarding the paper or the rebuttal. We are eager to embrace and integrate all your critiques into our work.

Thank you!

---

### Meta-Review · Area_Chair_kCxr · 2024-12-22

**Metareview:**

## Summary:
The paper develops a new benchmark for multimodal large language models (MLLMs) to measure the uncertainty and robustness of their answers to multi-choice questions when the input question is appended with explicit/implicit misleading instructions. They propose to generate the explicit misleading instructions by 12 templates, while the implicit ones are mainly generated by GPT-4o. They propose a misleading rate to measure the change of model responses from correct-to-incorrect and incorrect-to-correct. Experiments on 12 open-source LLMs and 5 close-source LLMs show that they suffer from high misleading rates on different misleading instructions. They further finetuned the open-source LLMs on the dataset and achieved a significant reduction in misleading rates.

## Strengths:
1. This paper studies an important topic regarding the robustness and uncertainty of MLLMs.
1. It is novel to study how to measure uncertainty under misleading instructions.
1. A new benchmark and two metrics that can enrich the evaluation of VLLMs.
1. Extensive experiments on multiple open-source and close-source models, covering explicit and implicit misleading scenarios.
1. The paper is very dense in details and results but better organization and highlights can significantly improve the presentation.

## Weaknesses:
1. It is not well justified that the misleading rating can faithfully and comprehensively measure the general uncertainty of LLM. It may only measure the uncertainty when the input contains conflicting information but not the uncertainty caused by the internal lack of knowledge or pitfalls of reasoning.
1. Reviewers find that the motivation of explicit misleading instructions might be questionable. Since the misleading input is a part of the whole instruction and the instruction-following capability is preferred in general, it is hard to justify whether the change of output answers is a preferred behavior or not. The challenge's definition might be ill-posed since following the misleading instruction and answering the question are conflicting in the scenario. Moreover, the explicit misleading instructions are generated by 12 templates for multi-choice questions, which are not sufficiently diverse to cover many misleading cases in practice.
1. Reviewers also raised several concerns regarding the implicit misleading. The paper and the following discussion do not provide sufficient information and examples (e.g., it is not clear what are the implicit instructions in Fig. 21). The original paper did not compare the implicit instructions generated by different models and humans. While new experimental results provided in the rebuttal are very helpful, there are still more details of the experiments that need to be clarified.
1. Reviewers raised several concerns about the finetuning and its induced improvement on MR. The improvement might not be surprising and trivial since the explicit misleading is not diverse (12 templates) and the patterns of implicit misleading generated by LLMs are too obvious. Evaluations on a more practical test set of misleading scenarios different from the templates and patterns in the finetuning set might be more convincing.
1. More experiments need to be presented, e.g., sampling different numbers of responses, open-ended questions, misleading instructions with different content/lengths/positions, MLLM with other modalities, etc. The authors responded with additional experiments, which can greatly improve the draft. Due to the time limit, these results are not entirely comprehensive. So another round of revision is necessary to make these new experiments more complete.

## Decision:
The authors provided detailed clarifications and various additional experimental results in the rebuttal. The reviewers share major concerns regarding the design of explicit/implicit instructions. To make the observations and claims more rigorous and convincing, various extra factors in the proposed setup whose impact on the model output and entanglement with the model uncertainty needed to be justified and removed if necessary. For this purpose, more comprehensive experiments in different settings are important. While the authors provided additional experiments requested by the reviewers, they were insufficient to resolve all the concerns.

Based on the above, the paper is not ready for publication yet. The meta-reviewer encourages the authors to further complete these experiments and simplify the problem setup of this paper, e.g., excluding the interference of other factors in the benchmark design and building a straightforward causal relation between the change of model outputs and the model uncertainty. How to measure MLLMs' uncertainty and robustness under misleading information is an important open challenge and the authors are encouraged to improve the study and prepare it for the next conference.

**Additional Comments On Reviewer Discussion:**

The authors provided detailed clarifications and various additional experimental results in the rebuttal. Two of the four reviewers responded to the rebuttal and confirmed that some major concerns have been addressed. The meta-reviewer carefully read the paper and all the discussions, especially the authors' responses to the two reviewers who have not responded to the discussion. The reviewers share major concerns regarding the design of explicit/implicit instructions. To make the observations and claims more rigorous and convincing, various extra factors in the proposed setup whose impact on the model output and entanglement with the model uncertainty needed to be justified and removed if necessary. For this purpose, more comprehensive experiments in different settings are important. While the authors provided additional experiments requested by the reviewers, they were insufficient to resolve all the concerns.

---

### Decision · Program_Chairs · 2025-01-22

Reject